# PSMA Radioligand Uptake as a Biomarker of Neoangiogenesis in Solid Tumours: Diagnostic or Theragnostic Factor?

**DOI:** 10.3390/cancers14164039

**Published:** 2022-08-21

**Authors:** Alessio Rizzo, Sara Dall’Armellina, Daniele Antonio Pizzuto, Germano Perotti, Luca Zagaria, Valerio Lanni, Giorgio Treglia, Manuela Racca, Salvatore Annunziata

**Affiliations:** 1Department of Nuclear Medicine, Candiolo Cancer Institute, FPO—IRCCS, 10060 Turin, Italy; 2Nuclear Medicine Unit, Department of Medical Sciences, AOU Città della Salute e della Scienza, University of Turin, 10134 Turin, Italy; 3Unità di Medicina Nucleare, TracerGLab, Dipartimento di Diagnostica per Immagini, Radioterapia Oncologica ed Ematologia, Fondazione Policlinico Universitario A. Gemelli, IRCCS, 00168 Rome, Italy; 4Imaging Institute of Southern Switzerland, Ente Ospedaliero Cantonale, 6501 Bellinzona, Switzerland; 5Faculty of Biology and Medicine, University of Lausanne, 1011 Lausanne, Switzerland; 6Faculty of Biomedical Sciences, Università della Svizzera Italiana, 6900 Lugano, Switzerland

**Keywords:** PSMA, PET, theragnostics, therapy, radioligand, nuclear medicine

## Abstract

**Simple Summary:**

Positron emission tomography/computed tomography (PET/CT) has an increasingly relevant role in the management of oncological patients. It consists of the administration of a radioactive molecule (tracer) which localizes in tumour cells due to specific cellular features (e.g., a receptor expressed by a specific cell population). PET/CT has a variety of applications in oncology, including staging, therapeutic response assessment and restaging in various types of tumours through the administration of different tracers. Among them, prostate-specific membrane antigen (PSMA) radioligands are a relatively novel compound widely employed in prostate cancer diagnostics and therapy. Besides their prostate cancer applications, PSMA radioligands have been shown to be a promising instrument to evaluate the stage and the aggressiveness in several types of cancer, since they allow us to study tumour neoangiogenesis. This review provides an overview of the applications of PSMA radioligand PET/CT in various types of tumours other than prostate cancer.

**Abstract:**

Due to its overexpression on the surface of prostate cancer cells, prostate-specific membrane antigen (PSMA) is a relatively novel effective target for molecular imaging and radioligand therapy (RLT) in prostate cancer. Recent studies reported that PSMA is expressed in the neovasculature of various types of cancer and regulates tumour cell invasion as well as tumour angiogenesis. Several authors explored the role of diagnostic and therapeutic PSMA radioligands in various malignancies. In this narrative review, we describe the current status of the literature on PSMA radioligands’ application in solid tumours other than prostate cancer to explore their potential role as diagnostic or therapeutic agents, with particular regard to the relevance of PSMA radioligand uptake as neoangiogenetic biomarker. Hence, a comprehensive review of the literature was performed to find relevant articles on the applications of PSMA radioligands in non-prostate solid tumours. Data on the general, methodological and clinical aspects of all included studies were collected. Forty full-text papers were selected for final review, 8 of which explored PSMA radioligand PET/CT performances in gliomas, 3 in salivary gland malignancies, 6 in thyroid cancer, 2 in breast cancer, 16 in renal cell carcinoma and 5 in hepatocellular carcinoma. In the included studies, PSMA radioligand PET showed promising performance in patients with non-prostate solid tumours. Further studies are needed to better define its potential role in oncological patients management, especially in those undergoing antineoangiogenic therapies, and to assess the efficacy of PSMA-RLT in this clinical context.

## 1. Introduction

Due to its overexpression on the surface of prostate cancer cells and its extracellular binding pocket for small-molecule ligands, glutamate carboxypeptidase II (GPII), also known as prostate-specific membrane antigen (PSMA), is a relatively novel effective target for molecular imaging and radioligand therapy (RLT) in prostate cancer [1]. These characteristics allow the development of small molecules binding PSMA, which are for all intents and purposes zinc-binding structures that interact with the binuclear zinc active site of GPII [2]. PSMA is a trans membrane protein encoded by the gene FOLH1 and was first discovered in prostate cancer cells in 1987 [3,4]. Conversely to what its name suggests, PSMA is not selective to prostate cancer cells, but is widely expressed in neovascular endothelial cells of various cancers (including brain, thyroid, renal, lung, liver, colo-rectal and breast) [5,6]. Recent pre-clinical studies suggest that GPII regulates tumour cell invasion and tumour angiogenesis by modulating integrin signal transduction in endothelial cells [7,8]. Considering the high expression of GPII on the cell membrane of prostate cancer cells and based on the first urea-based compounds designed by Kozikowski and colleagues in 2001 [9], several low-molecular weight radiolabelled GPII inhibitors have been developed with the purpose of improving the diagnostic performance of nuclear medicine imaging for prostate cancer.

To date, the most widely employed PSMA radioligand in clinical practice is ^68^Ga-PSMA-11 (also called ^68^Ga-DKFZ-PSMA-11 or ^68^Ga-PSMA-HBED-CC), developed by Eder and Haberkorn [10] and subsequently evaluated in clinical practice by Haberkorn and Afshar-Oromieh [11,12] at the German Cancer Research Centre in the University Hospital of Heidelberg, which consists of a Glu-urea-Lys inhibitor motif conjugated with Ga-specific acyclic chelator (HBED-CC) and shows higher affinity and internalization in prostate cancer cells than the corresponding DOTA analogue. Due to the overwhelming results obtained in the initial evaluation of ^68^Ga-PSMA-11, it was not long before the rapid development of therapeutic analogues and the first treatments with ^177^Lu-PSMA I&T (imaging and therapy), which employed DOTAGA as a chelator, and ^177^Lu-PSMA-617, a DOTA-conjugated compound [2]. Furthermore, DOTAGA- and DOTA-conjugated compounds can be labelled with other radionuclides such as ^90^Y for β^-^-therapy or ^225^Ac for α-therapy [13,14,15,16]. More recently, ^18^F-labeled PSMA radioligands were introduced in clinical practice: the main employed radiopharmaceuticals to date are ^18^F-DCFPyL and ^18^F-PSMA-1007 [17,18]. It is interesting that, conversely from any other PSMA radioligand, ^18^F-PSMA-1007 shows higher lipophilicity and, consequently, may not be found in the ureter and bladder since its clearance is liver-dominant [19]. In 2021 U.S. Food and Drug Administration (FDA) approved both ^68^Ga-PSMA-11 and ^18^F-DCFPyL administration in patients with suspected prostate cancer metastases who are potentially curable by surgery or radiation therapy and patients with suspected prostate cancer recurrence [20,21].

As reported by the phase III VISION trial [22], radioligand therapy with ^177^Lu-PSMA-617 was able to prolong progression-free survival (PFS) and overall survival (OS) when added to standard care in patients with advanced metastatic castration-resistant prostate cancer. ^177^Lu-PSMA-617 received FDA approval in March 2022 and its administration is indicated in patients with PSMA-positive metastatic castration-resistant prostate cancer previously treated with androgen-receptor pathway inhibitors and taxane-based chemotherapy [23]. This accomplishment makes us question whether similar results can be obtained by employing PSMA-RLT in other types of cancer expressing PSMA on tumour cells or the tumour-associated neovasculature. Angiogenesis is defined as a biological process in which new blood vessels are formed from pre-existing ones, leading to the development of the tumour’s blood supply, and it is considered to be a rate-limiting factor in cancer progression [24,25]. These findings were the basis for the development and the introduction into clinical practice of antineoangiogenic treatments [26]. The antineoangiogenic treatment rationale was to induce “tumour starvation”: preventing angiogenesis and nutrient supply, tumour progression should be slowed down. Since clinical studies with bevacizumab monotherapy did not show any benefit in terms of OS, this theory was soon abandoned [26]. This phenomenon might be explained by the concept of “vessel normalization”, in which vascular endothelial growth factor (VEGF) plays a key role, as VEGF-targeted therapies can restore the balance between anti- and pro-angiogenic factors, so the abnormal tumour vessels are remodelled into normal blood vessels which have a higher pO2, improved pericyte coverage, lower macromolecular permeability and improved delivery of therapeutic agents (this concept may be applied both for traditional chemotherapies and immunotherapies) [27,28]. These findings lead to a second question: can PSMA imaging predict which tumours are more eligible for antineoangiogenic treatment or predict its outcome?

The purpose of this narrative review is to assess whether PSMA radioligands, employed as diagnostic or theragnostic factors, might play a role in the management of patients with solid tumours other than prostate cancer, with particular regard to the significance of PSMA radioligand uptake as a neoangiogenesis biomarker.

## 2. Materials and Methods

### 2.1. Search Strategy

Three authors (A.R., G.T., S.A.) performed a comprehensive literature search from PubMed/MEDLINE, EMBASE and Cochrane library databases to find relevant published articles on the applications of PSMA radioligands in tumours other than prostate cancer. A search algorithm based on a combination of these terms and adapted to different organs was used: (“PET” OR “positron emission tomography”) AND (“PSMA” OR “DCFPyL”) AND (“cancer” OR “malignancy” OR “tumour”). Only articles in English were selected. No beginning date restriction was used. The literature search was last updated on 14th July 2022. To expand the research, references in the retrieved papers were also screened for additional studies. A literature search according to the Population, Intervention, Comparator, Outcomes (PICO) framework was performed, establishing criteria for study eligibility as follows: adult (≥18 years) patients with non-prostate solid tumour diagnosis (Population), undergoing PSMA radioligand PET or RLT (Intervention), with or without comparative imaging or therapy (Comparator); the considered outcomes were the evaluation of PSMA radioligand uptake in non-prostate tumour lesions, PSMA radioligand PET and therapy performance as well as change in patient management. All studies investigating the role or the application of PSMA radioligands to solid tumours other than prostate cancer were eligible for inclusion. Exclusion criteria were (a) articles not within the field of PET/CT or RLT, and (b) case reports, review articles, editorials, letters, comments, conference proceedings. Three researchers (A.R., G.T., S.A.) independently reviewed the titles and abstracts of the retrieved articles, applying the inclusion and exclusion criteria mentioned above and rejecting articles if they were clearly ineligible. The same three authors then independently reviewed the full-text version of the remaining articles to assess their eligibility for inclusion, resolving disagreements in a consensus meeting.

### 2.2. Data Extraction

For each eligible article, information was collected concerning the general study data, number of patients, employed radiopharmaceutical, investigated tumour, clinical setting, number of examined lesions, comparator exams. Moreover, the studies were divided into six groups depending on the primary tumour: (1) gliomas; (2) salivary glands malignancies; (3) thyroid cancer; (4) breast cancer; (5) renal cell cancer; and (6) hepatocellular carcinoma. Figure 1 summarizes the study selection process.

## 3. Results

### 3.1. Gliomas

Eight full-text articles assessing the potential role of PSMA radioligands in diagnostics of gliomas were reviewed [29,30,31,32,33,34,35,36]. Out of the eight included studies, three had a prospective design [30,32,34]. The sample size varied from 6 to 35 patients, and 178 patients with glioma were evaluated.

In all studies but one [29,30,31,32,33,34,36], PET/CT or PET/MRI was performed using ^68^Ga-PSMA-11, whereas the remaining study used ^68^Ga-PSMA-617 [35].

The main types of gliomas investigated were high-grade gliomas (e.g., glioblastoma). Regarding the clinical setting, PET using PSMA radioligands was performed in patients with gliomas to differentiate high-grade gliomas from low-grade gliomas (three articles [31,32,35]), to evaluate a suspicious recurrence of high-grade gliomas (three articles [33,34,36]), or for both indications (two articles [29,30]).

All of the examined papers used MRI as an comparator to evaluate PSMA radioligand PET/CT or PET/MRI performance [30,31,32,33,34,35,36], while three used also ^18^F-FDG PET/CT as a comparator [29,31,35].

Regarding the clinical findings, PET/CT with PSMA radioligands showed high diagnostic accuracy in detecting high-grade gliomas, both in discriminating high-grade gliomas from low-grade gliomas [29,30,31,32,35] and in detecting the recurrence of high-grade gliomas after treatment [29,30,33,34,36]. PSMA radioligand PET/CT showed comparable performances in high-grade gliomas when compared with MRI [32,34,36] and was superior to ^18^F-FDG PET both in staging and restaging gliomas mainly due to the absence of the high physiologic background in normal brain parenchyma in ^18^F-FDG PET/CT exams. In one study, immunohistochemistry analysis showed higher PSMA staining in high-grade gliomas [35]. Characteristics of the examined studies are reported in Table 1.

### 3.2. Salivary Glands Malignances

Three full-text papers assessing the potential role of PSMA radioligands in diagnostics of salivary glands cancer (SGC) were reviewed [37,38,39], which evaluated 9, 25 and 6 patients, respectively. Only one study had a prospective design [38]. All studies employed ^68^Ga-PSMA-11 as a radiopharmaceutical performing PET/CT. In one study, in addition to ^68^Ga-PSMA-11, ^177^Lu-PSMA-617 RLT was performed [39].

All of the included studies explored PSMA-targeted PET/CT performance in adenoid cystic carcinoma (AdCC) [37,38,39]. Other studied histotypes included salivary duct carcinoma (SDC) [38] and acinic cell carcinoma [39]. In all studies, PET/CT examination had the purpose of restaging patients with local or distant recurrence [37,38,39]. With regard to performance, in all studies PSMA-targeted PET/CT seemed to be a reliable instrument to detect local recurrences and distant metastases [37,38,39]. Regarding comparisons between different examinations, Klein Nulent and colleagues compared the performances of ^68^Ga-PSMA-11 PET/CT and ^18^F-FDG PET/CT, analysing 18 lesions in nine patients, of which 4 were local recurrences and 14 were distant metastases in different sites (meninx, lung, liver, bone), and reported quite similar performances between both tracers [37]; on the other hand, Van Boxtel compared ^68^Ga-PSMA-11 PET/CT with full-dose CT scan and reported that PSMA radioligand PET/CT had added diagnostic value in 4 AdCC patients out of the 15 included in the study; in particular, PSMA radioligand PET/CT was able to identify bone metastases in two patients, a AdCC local recurrence in the Bartholin gland and additional lymph node metastases not enhanced by CT examination [38].

All three of the included studies performed immunohistochemistry examination on AdCC primary tumours and metastases biopsies to evaluate PSMA staining and reported variable PSMA expression in cancer cells, ranging from <1% to 95% in local recurrences as well as distant metastases [37,38,39]. In this context, Van Boxtel evaluated PSMA staining in SDC and reported that PSMA is mainly expressed by the neovascular endothelium [38]. In one study, ^177^Lu-PSMA-617-RLT with palliative intent was performed in six patients with metastatic SGC [39]. Among the analysed patients, two completed the four cycles planned in the study protocol, achieving stable disease and partial response, respectively, three interrupted the RLT after two cycles due to disease progression, and the remaining one interrupted the treatment after one cycle due to the occurrence of side effects (fatigue, xerostomia). The characteristics of the included studies are reported in Table 2.

### 3.3. Thyroid Cancer

Six full-text papers assessing the potential role of PSMA radioligands in the diagnostics and therapy of differentiated thyroid cancer (DTC) were reviewed [40,41,42,43,44,45]. Out of the six included studies, four had a prospective design [40,42,43,45]. The sample size varied from 2 to 11 and from 3 to 43 for patient and lesion analyses, respectively.

In all studies but one [40,41,42,43,44], PET/CT was performed using ^68^Ga-PSMA-11, whereas the remaining study used ^18^F-DCFPyl [45]. In one study, in addition to ^68^Ga-PSMA-11 PET/CT as diagnostic compound, ^177^Lu-PSMA-617 RLT therapy was performed in two out of five patients [41].

The main DTC investigated histotype was papillary thyroid cancer, as overall a total of 27 patients were examined; further DTC subtypes included were follicular (10 patients), Hürtle cell (2 patients) and anaplastic carcinomas (2 patients). All of the available literature concerning the employment of PSMA radioligands in thyroid cancer focuses on radioiodine-refractory DTC patients [40,41,42,43,44,45].

Two out of the examined papers used only ^18^F-FDG PET/CT as a comparator to evaluate PSMA radioligand PET/CT performance [40,43], one used only an ^131^I whole post-therapeutic whole body scan in addition to SPECT/CT [44], two combined PSMA imaging with ^18^F-FDG PET/CT and ^123^I or ^131^I whole body scan [42,45] and the remaining paper did not compare PSMA results with any other exam [41]. When compared to ^18^F-FDG in radioiodine refractory thyroid cancer patients, PSMA radioligands showed conflicting results, since Lawn-Heath and colleagues reported the superior performance of ^18^F-FDG both in DTC patients and in dedifferentiated thyroid cancer patients [42], but other authors reported similar results between the two examinations or a slight superiority in favour of PSMA radioligand PET/CT [40,43,45]. Moreover, Pitalua-Cortes observed that ^68^Ga-PSMA-11 PET/CT had a superior capability for metastatic lesion detection when compared to ^131^I imaging in radioiodine refractory DTC patients [44].

None of the included studies reported immunohistochemistry data about PSMA staining in DTC.

With regard to clinical findings, PSMA radioligand PET/CT was shown to be a valuable instrument to restage DTC patients with thyroglobulin elevation and negative RAI scintigraphy, despites some evidence showing a heterogeneous uptake in different lesions [40,41,42,43]. The only study that evaluated ^177^Lu-PSMA-617-RLT performance after two cycles demonstrated a slight, temporary response in one patient and disease progression after one month in the other [41]; no side effects due to PSMA-RLT were reported. The characteristics of the examined studies are reported in Table 3.

### 3.4. Breast Cancer

Two full-text papers assessing the potential role of PSMA radioligands in the diagnostics of breast cancer (BC) were reviewed [46,47]. One of the examined studies was prospective [46], while the second had a retrospective design [47]. The number of evaluated patients was 19 and 15 in the two studies, which analysed 81 and 127 lesions, respectively [46,47]. Both the included studies reported heterogeneous PSMA radioligands uptake in different histopathological subtypes [46,47], with evidence of higher detection rates in luminal B and triple negative histologies [47]. When compared to ^18^F-FDG PET/CT, PSMA radioligand PET/CT showed similar performance in Her2-positive and triple negative subtypes, whereas poor performance was observed in luminal A and luminal B Her2-negative subtypes in primary tumour evaluation as well as in lymph node and distant metastases [47].

None of the included studies reported immunohistochemistry data about PSMA staining in BC. The characteristics of the included studies are reported in Table 4.

### 3.5. Renal Cell Carcinoma

Sixteen full-text papers assessing the potential role of PSMA radioligands in the diagnostics of renal cell carcinoma (RCC) were reviewed [48,49,50,51,52,53,54,55,56,57,58,59,60,61,62,63]. Out of the 16 included studies, 6 had a prospective design [48,49,52,53,57,59]. The sample size varied from 5 to 53 and from 22 to 94 for patient and lesion analyses, respectively.

In 10 studies, PET/CT was performed using ^68^Ga-PSMA-11 [49,50,51,54,55,57,59,60,61,62] while four used ^18^F-DCFPyl as a diagnostic radiopharmaceutical [48,52,53,56], one used ^18^F-PSMA-1007 [58], and the remaining study used both ^68^Ga-PSMA-11 and ^18^F-PSMA-1007 [63].

The main RCC histotype investigated was clear cell renal cell carcinoma (ccRCC), since a total of 278 patients were examined in 15 of the included papers [48,49,50,51,53,54,55,56,57,58,59,60,61,62,63]. Further RCC subtypes examined in the included studies were papillary (25 patients in 10 studies [49,50,51,52,54,58,59,60,61,62]) and chromophobe (14 patients in seven studies [50,52,54,59,60,61,62]); the number of patients with other subtypes of renal malignancies (e.g., oncocytoma, unclassified RCC) was 21 across nine studies [49,52,54,58,59,60,61,62,63]. With regard to the clinical setting in which PSMA radioligand imaging was employed, in eight studies, PET/CT had the purpose of staging untreated RCC patients [50,52,53,60,61,62], in two it had the purpose of restaging after surgery [56,57], in two it had the purpose of restaging after suspect findings in other radiological exams [48,52], while in one it had the purpose of assessing treatment response [58], and in the remaining three examined papers, both staging and restaging PET/CT exams were included [51,54,63].

Eight out of the examined papers used CT and/or MRI as a comparator to evaluate PSMA radioligand PET/CT performance [48,49,52,53,57,58,61,63], two used ^18^F-FDG PET/CT [51,56], while the remaining six performed only PSMA radioligand PET/CT as a diagnostic exam [50,54,55,59,60,62].

All studies evaluating PSMA radioligand PET/CT in ccRCC reported a good performance in assessing the presence of pathological findings in different anatomic sites, including the bone, brain, lymph nodes and soft tissues [48,49,50,51,53,54,55,56,57,58,59,60,61,62,63]. When compared to conventional imaging and ^18^F-FDG, PSMA radioligand PET/CT showed overall better performance in nine of the included studies which analysed ccRCC patients in all the examined clinical contexts [48,49,53,54,56,57,58,61,63]. Moreover, PSMA radioligand PET/CT was able to change management in 22 patients included in three studies [53,54,63]. Nevertheless, poorer performance was observed in non-ccRCC patients while comparing PSMA radioligand PET/CT to conventional imaging [52]. Based on these results, most of the authors warranted a potential role for this examination in all of the investigated clinical settings. In two papers, a correlation between negative prognostic histopathological factors such as the presence of VEGF receptor (VEGFR) 2/platelet derived growth factor receptor (PGDFR) β expression, tumour necrosis, sarcomatoid or rhabdoid features and ^68^GaPSMA-11 uptake as well as stronger PSMA staining in immunohistochemistry studies was found [55,60]. Conversely from what observed in ccRCC, PSMA-targeted PET/CT showed overall poor performance in other RCC subtypes when compared to conventional imaging [52]; moreover, weaker PSMA staining was found in RCC subtypes other than ccRCC [61]. The characteristics of the included studies are reported in Table 5.

### 3.6. Hepatocellular Carcinoma

Five full-text papers assessing the potential role of PSMA radioligands in the diagnostics of hepatocellular carcinoma (HCC) were reviewed [64,65,66,67,68]. Out of the five included studies, four had a prospective design [64,65,66,68]. The sample size varied from 7 to 40 and from 37 to 142 for patient and lesion analyses, respectively.

In all studies, PET/CT was performed using ^68^Ga-PSMA-11 as the diagnostic radiopharmaceutical [64,65,66,67,68].

With regard to the clinical setting in which PSMA radioligand imaging was employed, in all studies PET/CT had the purpose both to stage untreated HCC patients and to restage after at least one line of treatment (transarterial chemoembolization, surgery, radiofrequency or transarterial radioembolization) [64,65,66,67,68].

All the examined papers used CT and/or MRI as comparator to evaluate PSMA radioligand PET/CT performance [64,65,66,67,68] and two added ^18^F-FDG PET/CT in the analysis [64,66].

All studies evaluating PSMA radioligand PET/CT in HCC reported a good performance in assessing the presence of pathological findings in liver and lymph-nodes, representing a potential novel imaging modality for patients with HCC, especially for extrahepatic disease detection [64,65,66,67,68]. When compared to other examinations, PSMA radioligand PET/CT showed better performance than ^18^F-FDG PET/CT in two studies, both in liver lesion detection and extrahepatic involvement assessment (lymph nodes, bone, peritoneum) [64,66]; moreover, PSMA radioligand PET/CT showed quite similar performance compared to MRI, especially in liver lesion assessment [64,65,66,67,68].

Three out of the five included studies performed immunohistochemistry to assess PSMA staining in tumour tissue [64,65,68]. Among them, PSMA expression was reported in tumour-associated endothelial cells in all cases with rare concomitant intratumoral weak PSMA staining (5,5%) in canalicular HCC. Characteristics of the included studies are reported in Table 6.

## 4. Main Findings and Discussion

### 4.1. General

Since the introduction of PSMA radioligands in clinical practice for diagnostic or therapeutic purposes, several studies have evaluated their possible clinical applications in patients with solid tumours other than prostate cancer to improve the diagnostic accuracy or even to predict the outcome after different treatment schemes [29,30,31,32,33,34,35,36,37,38,39,40,41,42,43,44,45,46,47,48,49,50,51,52,53,54,55,56,57,58,59,60,61,62,63,64,65,66,67,68]. The in vivo study of biologic processes such as neoangiogenesis and hypoxia in tumour lesions represents a milestone in the history of cancer care, seeing as these features are targets of novel therapies [69]. Despite the rationale is consistent with a potential role for PSMA radioligands in this clinical context, the current status of the literature is still far from fulfilling this purpose, as we found the majority of extracted studies to be methodologically underpowered by sample size limitations. Moreover, a novel diagnostic examination is considered as being able to change patient management only if its employment might bring about an up- or down-staging when compared to conventional imaging and subsequently affects the treatment plan; despite recent evidences made clear that PSMA radioligand PET is able to change management in prostate cancer patients, this aspect needs to be furtherly explored in patients with tumours other than prostate cancer. Even so, this review tried to assess PSMA radioligands performance in patients with non-prostate solid tumours and whether these malignancies are potentially suitable for PSMA-targeted RLT. Regarding general characteristics, a common limitation, undeniable in most of the examined studies, was the limited number of included patients. The validation of a new employment for a radiopharmaceutical in a specific clinical context needs to be extensively consistent with large prospective cohorts and should rely on long-term follow-ups to assess whether the evidenced predictive values of PSMA radioligand PET/CT are confirmed and to assess if an eventual change in patient management due to PSMA-targeted PET/CT results actually brought benefits in terms of PFS and OS.

With regard to the employed radiopharmaceuticals, most of the included studies used ^68^Ga-PSMA-11 PET/CT as diagnostic compound to explore its potential role in several solid tumours other than prostate cancer [29,30,31,32,33,34,36,37,38,39,40,41,42,43,44,46,47,49,51,54,55,57,59,60,61,62,63,64,65,66,67,68]. To date, an increasing amount of PSMA radioligands is available for nuclear medicine physicians, including novel ^18^F-labelled radioligands [5]. Despite numerous studies comparing ^18^F-labeled and ^68^Ga-labeled PSMA radioligand performance finding similar results [70], using ^18^F-labelled compounds seems to rely on several advantages compared to using ^68^Ga-PSMA radioligands, since it can count on large-scale production, reduced costs, and high-quality images through lower positron energy, and a longer half-life [71]. However, since intense unspecific ^18^F-PSMA-1007 uptake may be observed within healthy bone, this tracer needs to be treated with caution when used for staging [72]. In this context, Alberts and colleagues presented a protocol design to experiment a head-to-head comparison of ^68^Ga-PSMA-11 and ^18^F-PSMA-1007 for the detection of recurrent prostate cancer lesions (ClinicalTrials.gov Identifier NCT05079828) [73]. As with prostate cancer, further studies on intra-patient comparisons with different PSMA-targeted tracers are needed for all of the clinical settings in which PSMA radioligands will find an application.

### 4.2. Gliomas

Gliomas are the most common primary tumours of the central nervous system, originating from the glial cells, and their annual incidence is about 6 per 100,000 cases worldwide [74,75]. Regarding grading, gliomas are most often referred to as low-grade gliomas or high-grade gliomas, based on the growth potential and aggressiveness of the tumours [76]. High-grade gliomas (e.g., glioblastomas) are characterized by a poor prognosis and high mortality, whereas low-grade gliomas present with a better prognosis [77]. Brain MRI is the diagnostic imaging modality of choice in the evaluation of patients with gliomas [75]. Different PET radiopharmaceuticals have been used to evaluate patients with gliomas with good diagnostic performances [78].

An excellent diagnostic performance of PSMA radioligand PET/CT or PET/MRI has been demonstrated before and after therapy in patients with high-grade gliomas (e.g., glioblastomas) [29,30,31,32,33,34,35,36]. These findings can be explained by the high expression of PSMA in the neovasculature of high-grade gliomas compared to low-grade gliomas or post-treatment abnormalities [79]. Notably, no significant uptake of PSMA radioligands was evident in the normal brain parenchyma, facilitating the detection of brain lesions with increased PSMA expression such as high-grade gliomas.

A clear advantage of PET with PSMA-targeting radiopharmaceuticals compared to ^18^F-FDG PET for detecting high-grade gliomas has been demonstrated [29,35,41]. The main limitation of ^18^F-FDG compared to PSMA radioligands is the very high physiological tracer uptake in the normal brain.

Overall, MRI remains the gold standard imaging method in the evaluation of gliomas [75], but PET with PSMA radioligands could be promising as a complementary imaging tool when MRI is doubtful. However, further studies should assess the clear diagnostic advantage of PSMA-targeted PET over MRI in high-grade gliomas.

The potential theragnostic role of PSMA radioligands could be the added value of PET with PSMA radioligands compared to PET with radiolabelled amino acids in high-grade gliomas. However, there are currently no studies comparing PET with these radiopharmaceuticals in high-grade gliomas. Furthermore, beyond case reports, the usefulness of PSMA-targeted therapy in patients with high-grade gliomas should be further demonstrated.

### 4.3. Salivary Gland Malignancies

SGCs are rare malignant head and neck malignancies, accounting for 3 to 10% of all head and neck tumours and exhibiting a varying clinical and biological behaviour; in this context, AdCC is one of the most common malignant SGCs, comprising 20–35% of all cases [80,81]. Its treatment usually consists of surgery and/or external beam radiation therapy in patients with local disease. Patients with locally advanced disease or distant metastases (frequently in bones or lungs) often show disease recurrence. In these patients, the effectiveness of both systemic chemotherapy and targeted immunotherapy is limited for symptomatic recurrent or distant disease, since a limited number of patients might show good response [82,83].

To date, ^18^F–FDG PET/CT plays a major role in the detection and staging of patients with head and neck cancer, with most studies focusing on squamous cell carcinoma (SCC). However, it is well known that AdCC has different biological characteristics, and its FDG-uptake is lower when compared to SCC [84]. In the recent literature, ^18^F-FDG PET/CT showed similar sensitivity for primary lesion detection when compared to CT and superior detection rates in lymph nodes and distant metastases [84].

With regard to PSMA radioligands physiologic biodistribution, it is well known that salivary glands show high uptake of PSMA radioligands [85]; nevertheless, our knowledge is poor about the mechanism underlying PSMA radioligand accumulation in salivary glands, since several immunochemistry studies observed the absence of PSMA expression in this district [86,87]. Moreover, PSMA-specific radioimmunoconjugates, including ^111^In-J591 and ^177^Lu-J591, did not show significant accumulation in healthy salivary glands [88]. These findings are consistent with the hypothesis supporting that PSMA radioligands uptake in salivary gland tissue is mainly unspecific.

Conversely from what was observed in the majority of cancer types included in this review, intracellular PSMA expression was observed in AdCC cells as well as in the tumour-induced neovasculature endothelium, although it seems not to be correlated with PSMA radioligand uptake in PET/CT images [37].

Concerning the clinical findings enhanced by the evaluated studies, PSMA radioligand PET/CT showed good performance while detecting and visualizing local recurrent and distant metastatic AdCC. Moreover, PSMA radioligand uptake was supported by high PSMA expression on immunohistochemistry studies [38].

With regard to the potential application of α- or β-emitter-radiolabeled PSMA-targeted ligands, only one study reported the employment of ^177^Lu-PSMA-617 for palliative intent in six patients, revealing partial response or stable disease in one-third of the enrolled patients [39]. Based on these premises and on the detection of PSMA staining in cancer cells, RLT with PSMA-targeted radioligands might be an option in AdCC patients when regular treatment options fail. In this context, more prospective studies are needed to assess whether RLT may improve prognosis in AdCC patients.

### 4.4. Thyroid Cancer

Thyroid cancer is the most common endocrine malignancy [89]. Despite its increasing incidence worldwide, most cases of DTC show excellent prognosis after surgery, usually concerning total or partial thyroidectomy with or without lymph-node dissection, TSH suppression due to levothyroxine treatment and radioiodine (RAI) therapy when indicated [89]. DTCs include papillary thyroid cancer (PTC) and follicular thyroid carcinoma (FTC); other less frequent subtypes of thyroid cancer are represented by medullary and anaplastic thyroid carcinomas, which usually show poor prognosis [89]. Nevertheless, about one-third of metastatic DTC patients do not demonstrate RAI uptake in their lesions or show progressive disease after RAI treatment and are classified as radioiodine-refractory DTC [89]. Most radioiodine-refractory DTCs show an aggressive histopathological feature and ^18^F-fluorodeoxyglucose (^18^F-FDG) uptake on PET/CT images [90,91].

Recent studies observed PSMA expression in the neovasculature of several thyroid cancer subtypes, including PTC and FTC, and revealed that strong PSMA staining tumours were more clinically aggressive than those with weak PSMA staining [92,93]. However, based on their findings, GPII is only expressed by tumour-associated vascular endothelium and not by cancer cells themselves. Moreover, it has been observed that PSMA expression may depend on DTC histotype: moderate- to high-grade PSMA staining has been observed in papillary and follicular subtypes and weak PSMA expression in anaplastic histologies.

As it is the main concern in physicians who deal with DTC, all the included studies enrolled patients with RAI-refractory thyroid cancer [40,41,42,43,44,45]. Since there are no reliable instruments to predict RAI-refractoriness and few lines of therapy are currently available in this clinical setting [94], this topic raised the attention of many authors searching for prognostic factors and new therapeutic agents.

With regard to clinical findings, PSMA radioligand PET/CT in DTC patients showed different uptake characteristics between primary recurrent lesions and distant metastases as well as among the included DTC histopathological subtypes [45].

Angiogenesis is a key step in tumour progression, and RAI-refractory DTC novel therapies with tyrosine kinase inhibitors (TKI), such as lenvatinib, operate by inhibiting the pathways through VEGFR, fibroblast growth factor receptor (FGFR), PGDFRα, rearranged during transfection (RET) and stem cell factor receptor (c-KIT), showing significant prolongation of PFS and improved response rate compared to placebo [94]. Based on these premises, PSMA radioligand PET/CT may have a role in predicting PFS and OS in RAI-refractory DTC patients undergoing TKI treatment.

Regarding PSMA-RLT, only one of the examined papers evaluated the employment of ^177^Lu-PSMA-617 in DTC patients, reporting a slight, temporary response in one patient [41]. The absence of GPII expression on the surface of tumour cells and its variable expression in neovasculature endothelium might explain why RLT showed low effectiveness in RAI-refractory DTC. More prospective studies with larger sample size are needed to correctly evaluate the actual effectiveness of PSMA-targeted RLT. A case of accidental finding of primary DTC is reported in Figure 2.

### 4.5. Breast Cancer

BC is the most common malignant tumour in women, with an estimated 2.3 million new cases per year and approximately 685,000 deaths per year worldwide [95]. This malignancy comprises various subtypes depending on its stage at presentation, histotype, biomarkers expression including estrogen receptor (ER), progesterone receptor (PgR) and human epidermal growth factor receptor 2 (Her2) as well as proliferation index (Ki67) [96,97,98]. Molecular and histological profiling of breast cancer is fundamental in treatment choice and is classified as: luminal A, characterized by ER and/or PgR expression and a Her2-negative arrangement, luminal B, characterized by ER and/or PgR expression in addition to a Her2-positive arrangement, the Her2-enriched subtype, characterized by the lack of ER and PgR expression but positive Her2 arrangement, and basal-like/triple-negative breast cancer, in which ER, PgR and Her2 are not expressed [96,97,98].

Despite BC demonstrating variable ^18^F-FDG avidity, ^18^F-FDG PET/CT plays a major role in BC management, especially in the staging of locally advanced BC, in restaging patients with suspect local disease or distant metastases recurrence, and in evaluating treatment response [99].

Tolkach et al. reported PSMA expression in the tumour-associated neovasculature endothelium in 315 cases of invasive non-special type BC and lobular BC [100]. Moreover, a stronger PSMA staining in higher-grade tumours, in Her2-positive BC and in tumours without ER and PgR expression was observed.

Two studies included in this review explored a possible role for PSMA radioligand PET/CT in staging BC and in restaging patients with suspect disease recurrence. Both studies reported a variable PSMA expression within the investigated lesions and higher uptake in Her2-positive BC subtypes and triple negative BC [46,47].

To date, only one case of ^177^Lu-PSMA-617 RLT in BC has been reported in a 38-year-old woman with triple negative BC with rapid progressive disease after multiple lines of systemic therapy. Since the patient showed further disease progression after two PSMA-targeted RLT administrations, treatment was dismissed [100]. However, considering the low toxicity of PSMA-targeted RLT, the authors warranted more studies to assess its efficacy in BC, especially in cases of triple negative subtypes.

### 4.6. Renal Cell Carcinoma

RCC is the most common type of renal tumour. Its incidence has progressively increased in recent decades due to a higher exposure to risk factors (e.g., obesity and alcohol consumption) and the development of more sensitive diagnostic modalities [101]. Based on its histopathology, RCC can be classified in two subtypes: ccRCC and non-ccRCC (which includes at least 15 histotypes, including papillary RCC and chromophobe RCC) [102]. About 20–30% of RCCs are present with distant metastatic lesions at diagnosis.

To date, contrast-enhanced CT and MRI are the main imaging methods employed to stage RCC, while ^18^F-FDG PET/CT has a limited role in its management, as the physiological renal excretion of ^18^F-FDG and metabolites hinders the characterization of primary tumours, and the expression of several enzymes such as fructose 1,6-bisphosphatase 1 seems to have an inverse correlation with ^18^F-FDG avidity in ccRCC [103,104].

Baccala and colleagues evaluated PSMA expression in a kidney tissue microarray including 169 samples of normal parenchyma and several cancer tissues, discovering a significant difference among endothelium of neovasculature in different RCCs [105]. In particular, the subtype with the greatest expression of PSMA was found to be ccRCC. GPII expression was also confirmed in the subtypes of chromophobe RCC and in oncocytoma, but not in papillary RCC, despite it being a neoplasm derived from proximal renal tubular cells, where PSMA is physiologically expressed [105,106]. Conversely from what reported by Baccala et al., a subsequent study from Al-Ahmadie and colleagues demonstrated PSMA expression in 75 RCC samples, including papillary RCC [107]. In particular, a direct relationship was found between the extension of neovascularization and the expression of PSMA, the latter being more strongly expressed in ccRCCa, known to be highly vascularized and expressed with less intensity and with a focal pattern in the papillary RCC, which showed poor vascularization. Moreover, Chang et al. demonstrated the expression of PSMA also in the neovasculature of RCC metastases as well as in the neovasculature endothelium cells of primary RCC [108].

PSMA radioligand PET imaging has been applied to detecting local recurrence and metastatic disease in RCC patients [48,51,52,54,56,63]. Moreover, one study preliminarily revealed that PSMA-based PET imaging demonstrated more precise response assessment and therapy monitoring of metastatic RCC patients to systemic treatment, usually consisting of TKIs or immune check-point inhibitors [58]. Several included studies tried to compare PSMA radioligand uptake characteristics with histopathologic features, reporting a correlation between intense ^68^Ga-PSMA-11 uptake and VEGFR-2/PDGFR-β expression and hypoxia-inducible factor (HIF) 2α [55,60].

No available studies were found concerning a possible employment of α- or β-emitter-radiolabelled PSMA-targeted ligands. A case of accidental finding of primary ccRCC is reported in Figure 3.

### 4.7. Hepatocellular Carcinoma

HCC is the third most common cause of cancer death worldwide, and its main risk factors are cirrhosis and viral hepatitis infection [109]. Despite several local and systemic treatments currently being available, treatment options for HCC may be limited at presentation due to the patient’s and tumour’s baseline status (lesions size, number and distribution, overall patient performance status), making aggressive management unsafe [110].

To date, HCC is diagnosed by imaging using CT or MRI based on Liver Imaging and Reporting Data System (LI-RADS) criteria [111]. Nevertheless, morphologic imaging is not able to explore tumour biology, usually characterized by marker molecular heterogeneity in this context. ^18^F-FDG PET/CT has sub-optimal liver imaging kinetics, since only high-grade HCC show significant uptake [112].

As for the majority of the solid tumours investigated in this review, immunohistochemistry studies reported PSMA staining only in the tumour-associated neovasculature endothelium [113]. Consistently with what was reported for other neoplasms, strong PSMA expression was associated with poor prognosis in patients with HCC [113]. However, it is interesting that in one paper, mild PSMA staining in canalicular HCC was described alongside tumour-associated neovasculature PSMA expression [114].

With regard to the clinical setting in which PSMA radioligand PET/CT was employed, the included studies explored potential applications of ^68^Ga-PSMA-11 both in staging and restaging for local or distant recurrence in patients previously treated with local or systemic therapies [64,65,66,67,68]. Overall, the included studies revealed a superior performance of PSMA-targeted PET/CT compared to ^18^F-FDG PET/CT and assessed that it might be a potential novel imaging modality supportive to MRI for local disease staging and restaging and to demonstrate extrahepatic involvement [67].

Since first-line therapy in locally advanced and metastatic HCC consists of a combination of immunotherapy and antineaongiogenic therapy (atezolizumab/bevacizumab) [115], the potential role of PSMA radioligands as a prognosis predictor deserves further analyses.

No available studies were found concerning a possible employment of α- or β-emitter-radiolabelled PSMA-targeted ligands.

### 4.8. Limitations

This review has some limitations. First, the available literature presents a limited amount of studies and most of them suffered from a poor number of enrolled patients undergoing PSMA radioligand administration for PET/CT diagnostics or RLT. Second, due to the heterogeneity of the studies, only a qualitative review with no quantitative meta-analytic data was considered for this literature review.

## 5. Conclusions

The exploration of PSMA radioligand PET/CT as a possible diagnostic or theragnostic agent in non-prostate solid tumours is in its infancy.

Since PSMA expression in solid tumours other than prostate cancer is primarily observed in the tumour neovasculature, and it showed promising performance compared to standard morphological and functional imaging, further studies are needed to assess whether this novel imaging examination might improve oncological patients’ management in specifical clinical settings such as the monitoring of antineoangiogenic therapies. As several authors pointed out [116], a subset of patients with non-prostate solid tumours show sufficient PSMA radioligand uptake in PET images to be potentially eligible for PSMA-RLT. In this context, further studies are needed to explore RLT feasibility and to assess if the absence of PSMA on cancer cell surface might be a rate-limiting factor for its success aside from the uptake detected in PET/CT images.

## Figures and Tables

**Figure 1 cancers-14-04039-f001:**
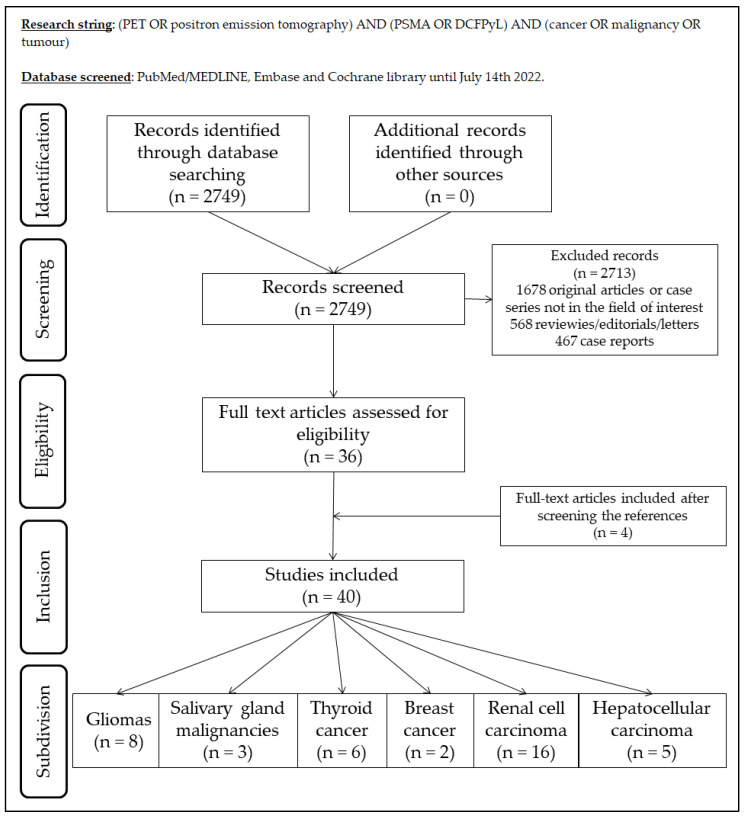
Summary of study selection process for the review.

**Figure 2 cancers-14-04039-f002:**
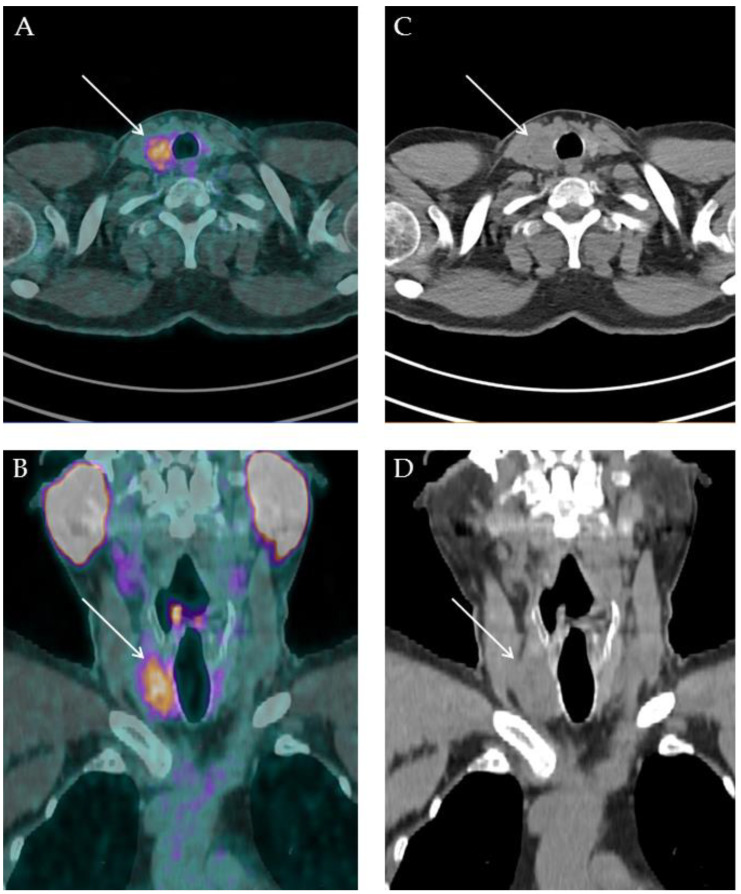
A 57-years old man with diagnosis of prostate adenocarcinoma previously treated through external beam radiation therapy in 2019 (Gleason score 4 + 4) came to our attention in December 2021 for biochemical recurrence (PSA value: 0.91 ng/mL) and underwent restaging ^18^F-PSMA-1007 PET/CT. PET/CT scan was performed 90 min after 270 MBq ^18^F-PSMA-1007 administration. PET/CT images showed focal uptake in the prostate gland right lobe, attributable to prostate cancer relapse. Furthermore, in the head and neck sections ((**A**) fused PET/CT axial, (**B**) fused PET/CT coronal, (**C**) low-dose CT axial, (**D**) low-dose CT coronal), an area of abnormal and intense ^18^F-PSMA-1007 uptake was observed in a thyroid nodule located in the right lobe. After routine diagnostic work up, the patient underwent total thyroidectomy without lymph-node dissection, and histologic examination was consistent with the diagnosis of tall cell variant papillary thyroid cancer.

**Figure 3 cancers-14-04039-f003:**
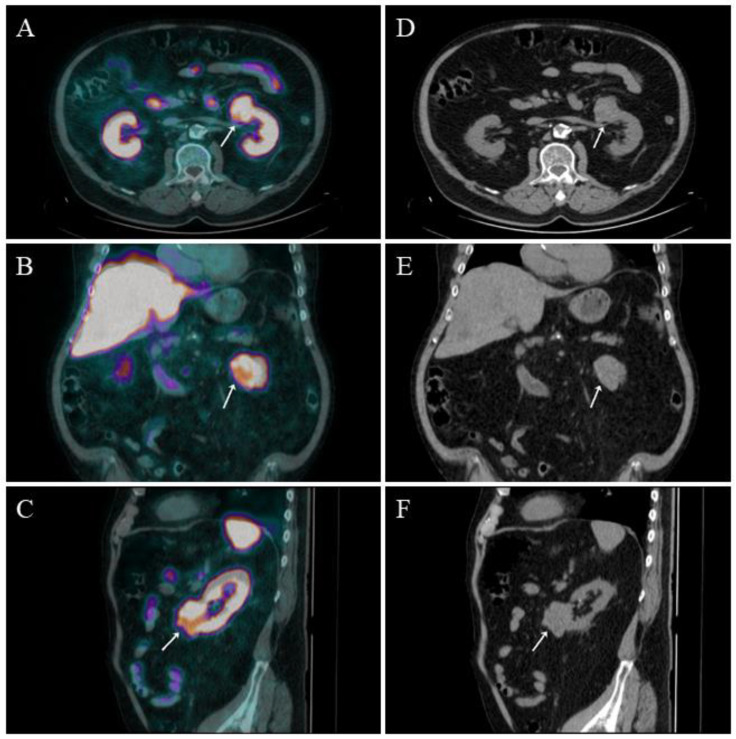
A 64-years old man with diagnosis of prostate adenocarcinoma previously treated through robot-assisted radical prostatectomy and pelvic lymph-node dissection in 2016 (pT2apN0c; Gleason score 4 + 3) came to our attention in February 2022 for biochemical recurrence (PSA value: 0.64 ng/mL) and underwent ^18^F-PSMA-1007 PET/CT. PET/CT scan was performed 90 min after 250 MBq ^18^F-PSMA-1007 administration. PET/CT images did not show any sign of suspect local recurrence nor distant metastases distinctly attributable to prostate cancer. Nevertheless, in the upper abdomen sections ((**A**) fused PET/CT axial, (**B**) fused PET/CT coronal, (**C**) fused PET/CT sagittal, (**D**) low-dose CT axial, (**E**) low-dose CT coronal; (**F**): low-dose CT sagittal), an area of abnormal and intense ^18^F-PSMA-1007 uptake was observed in an exophitic lesion located at the lower pole of the left kidney (white arrow). After routine diagnostic work up, the patient was submitted to total nephrectomy and histologic examination was consistent with the diagnosis of ccRCC.

**Table 1 cancers-14-04039-t001:** Characteristics of the included studies which employed PSMA radioligands in gliomas.

First Author and Year	Type	Country	N. Patients	Tracer	Histopathological Subtype (N. Patients)	Clinical Setting (N. Patients)	Analysed Lesions	Comparator
Sasikumar 2017 [29]	Not reported	India	6	^68^Ga-PSMA-11 (I)	6 high-grade	Initial diagnosis or restaging	6	MRI ^18^F-FDG PET/CT
Sasikumar 2018 [30]	Prospective	India	15	^68^Ga-PSMA-11 (I)	1 low-grade; 14 high-grade	Initial diagnosis or restaging	15	MRI
Verma 2019 [31]	Not reported	India	10	^68^Ga-PSMA-11 (I)	3 low-grade; 7 high-grade	Initial diagnosis	10	MRI ^18^F-FDG PET/CT
Akgun 2020 [32]	Prospective	Turkey	35	^68^Ga-PSMA-11 (I)	14 low-grade; 21 high-grade	Initial diagnosis	35	MRI
Kunikowska 2020 [33]	Not reported	Poland	15	^68^Ga-PSMA-11 (I)	15 high-grade	Restaging	15	MRI
Kumar 2021 [34]	Prospective	India	33	^68^Ga-PSMA-11 (I)	33 high-grade	Restaging	33	MRI
Liu 2021 [35]	Retrospective	China	30	^68^Ga-PSMA-617(I)	14 low-grade; 16 high-grade	Initial diagnosis	30	MRI ^18^F-FDG PET/CT
Kunikowska 2022 [36]	Not reported	Poland	34	^68^Ga-PSMA-11 (I)	34 high-grade	Restaging	34	MRI

PET: positron emission tomography, CT: computed tomography, MRI: magnetic resonance imaging, PSMA: prostate-specific membrane antigen, FDG: fluorodeoxyglucose, I: imaging.

**Table 2 cancers-14-04039-t002:** Characteristics of the included studies which employed PSMA radioligands in salivary gland malignancies.

First Author and Year	Type	Country	N. Patients	Tracer	Histopathological Subtype (N. Patients)	Clinical Setting (N. Patients)	Comparator
Klein Nulent 2017 [37]	Retrospective	Netherlands	9	^68^Ga-PSMA-11 (I)	AdCC	Restaging	^18^F-FDG PET/CT CT
Van Boxtel 2020 [38]	Prospective	Netherlands	25	^68^Ga-PSMA-11 (I)	15 AdCC 10 SDC	Restaging	CT
Klein Nulent 2021 [39]	Retrospective	Netherlands	6	^68^Ga-PSMA-11 (I) ^177^Lu-PSMA-617 (T)	4 AdCC 1 unclassified adenocarcoma 1 acinic cell carcinoma	Palliative RLT	/

PET: positron emission tomography, CT: computed tomography, PSMA: prostate-specific membrane antigen, FDG: fluorodeoxyglucose, RLT: radioligand therapy, AdCC: adenoid-cystic carcinoma, SDC: salivary duct carcinoma, I: imaging, T: therapy.

**Table 3 cancers-14-04039-t003:** Characteristics of the included studies which employed PSMA radioligands in thyroid cancer.

First Author and Year	Type	Country	N. Patients	Tracer	Histopathological Subtype (N. Patients)	Clinical Setting	Analysed Lesions	Comparator
Lütje 2017 [40]	Prospective	Germany	6	^68^Ga- PSMA-11 (I)	2 papillary 4 follicular	Radioiodine refractory TC	42	^18^F-FDG PET/CT
De Vries 2020 [41]	Retrospective	Netherlands	5	^68^Ga-PSMA-11 (I) ^177^Lu-PSMA-617 (T)	4 papillary 1 follicular variant	Radioiodine refractory TC	/	/
Lawhn-Heath 2020 [42]	Prospective	USA	11	^68^Ga- PSMA-11 (I)	3 papillary 2 follicular 2 Hurthle cell 2 poorly differentiated 2 anaplastic	TC with abnormal uptake on ^18^F-FDG PET and/or ^123^I /^131^I scan	43	^18^F-FDG PET/CT ^123^I /^131^I scan
Verma 2021 [43]	Prospective	India	9	^68^Ga-PSMA-11 (I)	7 papillary 2 follicular variant	Radioiodine refractory TC	14	^18^F-FDG PET/CT
Pitalua-Cortes 2021 [44]	Retrospective	Mexico	10	^68^Ga- PSMA-11 (I)	7 papillary 3 follicular	Radioiodine refractory TC	64	^131^I scan
Santhanam 2021 [45]	Prospective	USA	2	^18^F-DCFPyl (I)	1 papillary 1 follicular	Radioiodine refractory TC	2	^18^F-FDG PET/CT ^123^I scan

PET: positron emission tomography, CT: computed tomography, PSMA: prostate-specific membrane antigen, DCFPyL: piflufolastat, FDG: fluorodeoxyglucose, TC: thyroid cancer, TKI: tyrosine-kinase inhibitor, I: imaging, T: therapy.

**Table 4 cancers-14-04039-t004:** Characteristics of the included studies which employed PSMA radioligands in breast cancer.

First Author and Year	Type	Country	N. Patients	Tracer	Histopathological SubType (N. Patients)	Clinical Setting (N. Patients)	Analysed Lesions	Comparator
Sathekge 2017 [46]	Prospective	South Africa	19	^68^Ga-PSMA-11 (I)	13 ductal 2 lobular 1 neuroendocrine differentiation 3 unknown	9 staging 5 restaging for local recurrence 5 restaging for distant metastases	81	CT bone scan ^18^F-FDG PET/CT
Medina-Ornelas 2020 [47]	Retrospective	Mexico	21	^68^Ga-PSMA-11 (I)	4 luminal A 4 luminal B Her2+ 2 luminal B Her2- 5 triple negative	Staging of locally advanced and metastatic BC	127	^18^F-FDG PET/CT

PET: positron emission tomography, CT: computed tomography, PSMA: prostate-specific membrane antigen, FDG: fluorodeoxyglucose, I: imaging.

**Table 5 cancers-14-04039-t005:** Characteristics of the included studies which employed PSMA radioligands in renal cell carcinoma.

First Author and Year	Type	Country	N. Patients	Tracer	Histopathological Subtype (N. Patients)	Clinical Setting (N. Patients)	Analysed Lesions	Comparator
Rowe 2015 [48]	Prospective	USA	5	^18^F-DCFPyL (I)	5 clear cell	Restaging in metastatic patients naive to systemic therapies	29	CT and MRI
Rhee 2016 [49]	Prospective	Australia	10	^68^Ga-PSMA-11 (I)	8 clear cell 1 papillary 1 unclassified	Staging in metastatic patients	86	CT
Sawicki 2016 [50]	Retrospective	Germany	6	^68^Ga-PSMA-11 (I)	4 clear cell 1 papillary 1 chromophobe	Staging in metastatic patients	22	/
Siva 2017 [51]	Retrospective	Australia	8	^68^Ga-PSMA-11 (I)	7clear cell 1 papillary	2 staging 5 restaging for suspect recurrence 2 staging and restaging after therapy		^18^F-FDG PET/CT
Yin 2018 [52]	Prospective	USA	8	^18^F-DCFPyL (I)	3 papillary 2 chromophobe 2 unclassified 1 xp11 translocation	Restaging in metastatic patients previously treated with multiple lines of systemic therapy	73	CT MRI
Meyer 2019 [53]	Prospective	USA	14	^18^F-DCFPyL (I)	14 clear cell	Staging in oligometastatic patients	47	CT MRI
Raveenthiran 2019 [54]	Retrospective	Australia	38	^68^Ga-PSMA-11 (I)	28 clear cell 1 oncocytoma 1 papillary 1 chromophobe 1 TCC 6 unknown	16 primary staging 32 restaging of suspected recurrent disease	51	/
Gao 2020 [55]	Retrospective	China	36	^68^Ga-PSMA-11 (I)	36 clear cell	Untreated primary renal cell carcinoma	/	/
Liu 2020 [56]	Retrospective	China	15	^18^F-DCFPyL (I)	15 clear cell	Post-operative restaging	42	^18^F-FDG PET/CT
Gühne 2021 [57]	Prospective	Germany	8	^68^Ga-PSMA-11 (I)	8 clear cell	Post-operative restaging	12	CT
Mittlemeier 2021 [58]	Retrospective	Germany	11	^18^F-PSMA-1007 (I)	8 clear cell 2 papillary 1 undifferentiated	Response assessment after therapy	/	CT
Golan 2021 [59]	Prospective	Israel	29	^68^Ga-PSMA-11 (I)	18 clear cell 4 papillary 2 chromophobe 2 oncocytoma 2 angiomyolipoma 1 mixed epithelial and stromal tumour	Dynamic PET/CT in the evaluation of localized renal masses.	29	/
Gao 2022 [60]	Retrospective	China	48	^68^Ga-PSMA-11 (I)	37 clear cell 4 papillary 3 chromophobe 4 unclassified	Staging and comparison between PET parameters with VEGFR-2/PDGFR-β expression	48	/
Li 2022 [61]	Retrospective	China	31	^68^Ga-PSMA-11 (I)	40 clear cell 3 papillary 1 chromophobe 1 mucinous 1 poorly differentiated 4 Others	Staging in metastatic patients	94	CT MRI
Meng 2022 [62]	Retrospective	China	53	^68^Ga-PSMA-11 (I)	40 clear cell 5 papillary 4 chromophobe 4 other	Staging	53	/
Tariq 2022 [63]	Retrospective	Australia	11	^68^Ga-PSMA-11 (I) ^18^F-PSMA-1007 (I)	10 clear cell 1 unclassified	4 staging 7 restaging	/	CT MRI

PET: positron emission tomography, CT: computed tomography, MRI: magnetic resonance imaging, PSMA: prostate-specific membrane antigen, DCFPyL: piflufolastat, FDG: fluorodeoxyglucose, TCC: transitional cell cancer, VEGFR: vascular endothelial growth factor receptor, PDGFR: platelet derived growth factor receptor, I: imaging, T: therapy.

**Table 6 cancers-14-04039-t006:** Characteristics of the included studies which employed PSMA radioligands in hepatocellular carcinoma.

First Author and Year	Type	Country	N. Patients	Tracer	Clinical Setting (N. Patients)	Analysed Lesions	Comparator
Kesler 2019 [64]	Prospective	Israel	7	^68^Ga-PSMA-11 (I)	6 staging 1 restaging after TACE	37	^18^F-FDG PET/CT CT MRI
Kunikowska 2021 [65]	Prospective	Poland	15	^68^Ga-PSMA-11 (I)	10 staging, 4 restaging after TACE, 1 restaging after hemiepatectomy and TACE	44	CT MRI
Gündoğan 2021 [66]	Prospective	Turkey	14	^68^Ga-PSMA-11 (I)	12 staging 1 restaging after TACE, 1 restaging after radiofrequency ablation + TACE.	61	^18^F-FDG PET/CT MRI
Hirmas 2021 [67]	Retrospective	Germany	40	^68^Ga-PSMA-11 (I)	27 Staging 13 restaging after local or systemic treatment.	142	CT
Thompson 2022 [68]	Prospective	USA	31	^68^Ga-PSMA-11 (I)	Staging	39	MRI

PET: positron emission tomography, CT: computed tomography, MRI: magnetic resonance imaging, PSMA: prostate-specific membrane antigen, DCFPyL: piflufolastat, FDG: fluorodeoxyglucose, TACE: transarterial chemo-embolization, TARE: transarterial radio-embolization, I: imaging.

## Data Availability

Data supporting reported results are available in public bibliographic databases (e.g., PubMed/Medline).

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
