# Peer review of "PSMA Radioligand Uptake as a Biomarker of Neoangiogenesis in Solid Tumours: Diagnostic or Theragnostic Factor?"

_cancers, 2022, doi:10.3390/cancers14164039_

Round 1

Reviewer 1 Report

This was an interesting read, and your review gives a good overview of the current data concerning the use of PSMA radioligands in tumors other than prostate cancer.

some comments:

·       It is not clear what kind of review you performed. Please indicate if narrative or systematic. If systemic please comment on whether you followed the PRISMA guidelines.

·       I would recommend using a flow chart to give a better overview of reviewed, in- and excluded studies.

·       Introduction:

o   The introduction is well written and gives comprehensive background information. However, I would recommend working again on the last part of the introduction. You could omit: “As already stated, PSMA is a transmembrane protein which is involved in tumour angiogenesis” (redundant). Also, you write that “these findings make arise a second question: can PSMA imaging predict which tumours are more eligible for antineoangiogenic treatment or predict its outcome?” whereby I could not find a first question that you would like to answer.

·       Materials & Methods:

o   Please provide your PICO(S) that you used to answer your research question

o   If so, please indicate the number if the protocol of your review was preregistered (e.g., in Prospero).

·       Results:

o   You state in the concluding part of the abstract that “PSMA-radioligands PET/CT showed promising performances while compared to standard morphological and functional imaging“. In the results section, you have six paragraphs (one for each cancer type) also including a table with study characteristics respectively. However, I miss the metric that you used to evaluate comparison performance (kappa metrics?). Moreover, it would be interesting how many lesions were identified in conventional and molecular imaging to see if there are differences (site, number). If your aim is to compare the performance of imaging modalities, you could consider excluding studies that have no comparator.

o   Also, you write in your abstract that the “Purpose of this review is to assess if PSMA-radioligands, employed as diagnostic or theragnostic factors, might play a role as prognosis predictor or therapeutic agent in solid tumours different from prostate cancer, with particular regard as neoangiogenetic biomarker”. Here, I miss the outcome variable how you would like to measure the prognostic value?

o   Table 1: Please review again the study design of the studies that you report that they do not indicate the study design. I checked ref32 which is retrospective.

o   Tables: Capitalize the C in “characteristics“ in the table headers. Moreover, you could omit the cancer type in the column “Histopathological Subtype“. Please indicate what the numbers mean in the columns histopath subtype and clinical setting in the table legends. Please also add a column with immunohistochemistry PSMA staining data (if available); and if the radioligand was used in therapy, please also add a column with adverse events that may have occurred.

o   A few typos and formatting issues: TKI: thytosine-kinase inhibitor, hystotype, double empty spaces, formatting the tables (e.g., Table 4 row Sathekge 2017 clinical setting, the number should be in the same line as the setting. Please also rephrase this sentence to make it easier to understand: “further RCC subtypes included were papillary (25 patients in 10 studies, chromophobe RCC (14 patients in 7 studies [47, 49, 51, 56-59]); other subtypes of renal malignancies explored (e.g. oncocytoma, unclassified RCC) were 21 in 9 studies. With regard to the clinical setting in which PSMA-radioligand imaging was employed, in 8 studies PET/CT had the purpose to stage untreated RCC patients, in 2 to restage after surgery (…)”. Please specify what the number 21 means here: “other subtypes of renal malignancies explored (e.g. oncocytoma, unclassified RCC) were 21 in 9 studies”.

·       Discussion:

o   Please discuss how differences between imaging modalities (if there are any) might affect treatment management.

Author Response

Thanks to the reviewer for the comments.

1. It is not clear what kind of review you performed. Please indicate if narrative or systematic. If systemic please comment on whether you followed the PRISMA guidelines.

We rephrased the sentences to make clear our review is a narrative review both in abstract and in introduction.

2. I would recommend using a flow chart to give a better overview of reviewed, in- and excluded studies.

We added a flow chart of in- and excluded papers.

3. The introduction is well written and gives comprehensive background information. However, I would recommend working again on the last part of the introduction. You could omit: “As already stated, PSMA is a transmembrane protein which is involved in tumour angiogenesis” (redundant). Also, you write that “these findings make arise a second question: can PSMA imaging predict which tumours are more eligible for antineoangiogenic treatment or predict its outcome?” whereby I could not find a first question that you would like to answer.

We rephrased the sentences as requested by the reviewer, moreover we added a comment to the sentence we intended as “first question”.

4. Please provide your PICO(S) that you used to answer your research question.

We added our PICO in the “Search strategy” subsection.

5. If so, please indicate the number if the protocol of your review was preregistered (e.g., in Prospero).

The proposed review was not preregistered.

6. You state in the concluding part of the abstract that “PSMA-radioligands PET/CT showed promising performances while compared to standard morphological and functional imaging“. In the results section, you have six paragraphs (one for each cancer type) also including a table with study characteristics respectively. However, I miss the metric that you used to evaluate comparison performance (kappa metrics?). Moreover, it would be interesting how many lesions were identified in conventional and molecular imaging to see if there are differences (site, number). If your aim is to compare the performance of imaging modalities, you could consider excluding studies that have no comparator.

We rephrased the abstract and added in each paragraph of results section the comparison between  different examination.

7. Also, you write in your abstract that the “Purpose of this review is to assess if PSMA-radioligands, employed as diagnostic or theragnostic factors, might play a role as prognosis predictor or therapeutic agent in solid tumours different from prostate cancer, with particular regard as neoangiogenetic biomarker”. Here, I miss the outcome variable how you would like to measure the prognostic value?

We reconsidered the purpose of the review since there are no studies assessing the prognostic value of PSMA-radioligand uptake and modified both abstract and text.

8. Table 1: Please review again the study design of the studies that you report that they do not indicate the study design. I checked ref32 which is retrospective.

We checked the study design of the papers included in table 1.

9. Tables: Capitalize the C in “characteristics“ in the table headers. Moreover, you could omit the cancer type in the column “Histopathological Subtype“. Please indicate what the numbers mean in the columns histopath subtype and clinical setting in the table legends. Please also add a column with immunohistochemistry PSMA staining data (if available); and if the radioligand was used in therapy, please also add a column with adverse events that may have occurred.

With the exception of the immunohistochemistry PSMA staining column, we modified the tables as requested by the reviewer. Nevertheless, we added PSMA staining data and RLT adverse events of the included studies in results sections.

10. A few typos and formatting issues: TKI: thytosine-kinase inhibitor, hystotype, double empty spaces, formatting the tables (e.g., Table 4 row Sathekge 2017 clinical setting, the number should be in the same line as the setting. Please also rephrase this sentence to make it easier to understand: “further RCC subtypes included were papillary (25 patients in 10 studies, chromophobe RCC (14 patients in 7 studies [47, 49, 51, 56-59]); other subtypes of renal malignancies explored (e.g. oncocytoma, unclassified RCC) were 21 in 9 studies. With regard to the clinical setting in which PSMA-radioligand imaging was employed, in 8 studies PET/CT had the purpose to stage untreated RCC patients, in 2 to restage after surgery (…)”. Please specify what the number 21 means here: “other subtypes of renal malignancies explored (e.g. oncocytoma, unclassified RCC) were 21 in 9 studies”.

We rephrased as requested.

11. Please discuss how differences between imaging modalities (if there are any) might affect treatment management.

We added a paragraph in discussion section.

Reviewer 2 Report

The manuscript by Rizzo et al. gives a comprehensive overview about the application of PSMA-radioligands for patients suffering from non-prostatic malignancies. Much effort has been put in the description of the analyzed clinical investigations, with respect to patient populations and clinical settings. Overall the application of the well-established PSMA ligands to solid tumors, other than prostatic lesions, is highly promising but needs further investigations, as also stated by the authors. The manuscript is well written and well structured. As a general remark, some PET-scans would be nice to illustrate the performance of the PSMA ligands in the described malignancies. Moreover, the work shows some content-related overlaps with the previously published review from Uijen et al. (PSMA radioligand therapy for solid tumors other than prostate cancer: background, opportunities, challenges, and first clinical reports, EJNMMI 2021), which however also included case reports, not included in the present study.

Please consider the following aspects:

-        PSMA radioligand as a biomarker. In my opinion this term is confusing. PSMA can be a biomarker, likewise PSMA-radioligand uptake. But a radioligand itself is not a biomarker. Please consider rephrasing (Title, line 36, line 112)

-        Please add the information that several of the ligands in these study have recently been approved by the FDA for prostate cancer imaging and therapy: 68Ga-PSMA-11, 177Lu-PSMA-617, 18F-DCFPyl

-        PSMA-1007: please add that unspecific bone uptake of this ligand needs to be treated with caution when used for staging

-        Section 3.2: what happened with the patients treated with Lu-PSMA-617

-        Section 3.4: results?

-        Line 295: comment of intrapatient comparisons. There are numerous studies evaluating 18F-PSMA ligands and comparing results to 68Ga-PSMA. Overall the clinical performance is quite similar, however the major advantage of 18F-PSMA lies on large-scale productions, reducing costs and feasibility of centralized productions.

-        Section 4.3: please add that healthy salivary glands show high PSMA radioligand uptake, which is at least in parts unspecific

-        Line 449: 18F is not excreted via the renal pathway. Please rephrase to 18F-FDG and metabolites

-        General remark: Add some figures of PSMA-PET-scans in patients with the described malignancies

Author Response

Thanks to the reviewer for the comments.

We are aware that the proposed review has some content-related overlaps with the previously published review from Uijen et al., but we wanted to make the reader think not only about the presence (or absence) of PSMA-radioligand uptake in different kind of tumours but also to understand that the mechanism underlying the uptake of PSMA-radioligands and the type of cells it accumulates in (cancer cells or neovascular endotherlium) might change its employment in clinical practice from diagnostic agent to theragnostic based on the kind of patient we are going to analyse.

1. PSMA radioligand as a biomarker. In my opinion this term is confusing. PSMA can be a biomarker, likewise PSMA-radioligand uptake. But a radioligand itself is not a biomarker. Please consider rephrasing (Title, line 36, line 112).

We rephrased as requested by the reviewer.

2. Please add the information that several of the ligands in these study have recently been approved by the FDA for prostate cancer imaging and therapy: 68Ga-PSMA-11, 177Lu-PSMA-617, 18F-DCFPyl.

We added FDA approval for the indicated radiopharmaceuticals in introduction section.

3. PSMA-1007: please add that unspecific bone uptake of this ligand needs to be treated with caution when used for staging.

We added 18F-PSMA-1007 unspecific bone uptake in discussion section.

4. Section 3.2: what happened with the patients treated with Lu-PSMA-617

We added data about patients who underwent RLT.

5. Section 3.4: results?

We added results in 3.4 section.

6. Line 295: comment of intrapatient comparisons. There are numerous studies evaluating 18F-PSMA ligands and comparing results to 68Ga-PSMA. Overall the clinical performance is quite similar, however the major advantage of 18F-PSMA lies on large-scale productions, reducing costs and feasibility of centralized productions.

We rephrased as requested by the reviewer.

7. Section 4.3: please add that healthy salivary glands show high PSMA radioligand uptake, which is at least in parts unspecific

We added physiologic salivary gland uptake in discussion section.

8. Line 449: 18F is not excreted via the renal pathway. Please rephrase to 18F-FDG and metabolites

Rephrased as requested.

9. General remark: Add some figures of PSMA-PET-scans in patients with the described malignancies

We added two figures about incidental findings in 18F-PSMA-1007 PET/CT of a thyroid cancer and a renal cell carcinoma.

Round 2

Reviewer 2 Report

The authors adressed all issues requested during the first review process.